# DistillCSE: Distilled Contrastive Learning for Sentence Embeddings[*]

**Jiahao Xu**[1]    **Wei Shao**[2]    **Lihui Chen**[1]    **Lemao Liu**[3]

[1]Nanyang Technological University, [2]City Univeristy of Hong Kong, [3]Tencent AI Lab

[1]jiahao004@e.ntu.edu.sg    elhchen@ntu.edu.sg
[2]weishao4-c@my.cityu.edu.hk
[3]redmondliu@tencent.com

## Abstract

This paper proposes the DistillCSE framework, which performs contrastive learning under the self-training paradigm with knowledge distillation. The potential advantage of DistillCSE is its self-enhancing feature: using a base model to provide additional supervision signals, a stronger model may be learned through knowledge distillation. However, the vanilla DistillCSE through the standard implementation of knowledge distillation only achieves marginal improvements due to severe overfitting. The further quantitative analyses demonstrate the reason that the standard knowledge distillation exhibits a relatively large variance of the teacher model's logits due to the essence of contrastive learning. To mitigate the issue induced by high variance, this paper accordingly proposed two simple yet effective solutions for knowledge distillation: a Group-P shuffling strategy as an implicit regularization and the averaging logits from multiple teacher components. Experiments on standard benchmarks demonstrate that the proposed DistillCSE outperforms many strong baseline methods and yields a new state-of-the-art performance.

## 1 Introduction

Sentence embedding aims to encode the sentence's semantic information from a discrete language space to continuous dense vectors which preserves the semantic meaning of the original sentences. It plays a central role in many downstream NLP tasks, for instance, document summarization (Gao et al., 2019), corpus mining (Bennani-Smires et al., 2018), and machine translation (Wang et al., 2017).

Recently, with the emergence of contrastive learning (CL), SimCSE (Gao et al., 2021) pioneers the current mainstream approaches of sentence embeddings. It combines CL with pre-trained language models (PLMs) (Devlin et al., 2019; Liu

et al., 2019) for training by pulling positive samples closer and pushing in-batch negatives apart, leading to state-of-the-art performance. Subsequently, a plethora of sentence embedding methods has been developed based on the SimCSE framework (Zhang et al., 2020; Yan et al., 2021; Giorgi et al., 2021; Kim et al., 2021; Carlsson et al., 2021; Zhou et al., 2022; Chuang et al., 2022; Clark et al., 2020; Dangovski et al., 2021; Zhang et al., 2022; Deng et al., 2019; Xu et al., 2023). Unfortunately, one problem with contrastive learning for sentence embeddings is that the construction of positive and negative sample pairs is often too simple, making it easy for the model to distinguish between positive and negative pairs. As a result, the model may not learn very informative knowledge, leading to sub-optimal performance (Tian et al., 2020b; Wang and Qi, 2022).

To this end, inspired by self-training (Yarowsky, 1995; scu, 1965), this paper proposes a framework –DistillCSE– which performs contrastive learning under the self-training paradigm with knowledge distillation (Hinton et al., 2015). The advantage of DistillCSE is its self-enhancing feature: using a base model to provide additional supervision signals, a stronger model can be learned through knowledge distillation. Specifically, our framework can be divided into three steps (§2): First, it learns a base model as a teacher using standard contrastive learning; Second, it learn a stronger student model through knowledge distillation (Gao et al., 2023); Thrid, it iteratively repeats the process of knowledge distillation by treating the student model as a teacher.

However, it is far from easy to put DistillCSE into practice: our preliminary experiment shows that the vanilla implementation of the proposed framework only achieves marginal improvements. We identify that the vanilla distillation method suffers from the severe overfitting on the training corpus (See Table 9 later). This motivates us to investi-

---

[*]The source code is available at https://github.com/Jiahao004/DistillCSE.

gate the in-depth reason why overfitting occurs. One possible intuition is that in the contrastive learning scenario, the logits of the teacher model are defined on a pair of examples; whereas in the usual scenario these logits are defined on a single example. This essential difference may lead to a relatively large variance of the teacher model's logits in the contrastive learning scenario. To demonstrate our insight, two metrics are designed to quantify the variance of logits: one variance measures the change of logits between training and testing examples which controls over-fitting, and the other measures the change of logits with respect to different teacher models. Through our quantitative analysis, it is observed that logits defined on an example pair indeed have a much larger variance than those defined on a single example (§3.1). To mitigate these two high-variance problems, we respectively proposed two simple yet effective solutions for the knowledge distillation step: a group-p shuffling strategy as an implicit regularization to prevent overfitting issue (§3.2) and the averaging logits from multiple teacher components to reduce the second variance w.r.t different teacher models (§3.3). Experiments on standard benchmarks demonstrate that the proposed DistillCSE outperforms many strong baseline methods and yields a new state-of-the-art performance (§4).

In summary, our contribution is three-fold:

1. We first pinpoint an important issue about knowledge distillation for contrastive learning: teacher logits exhibit high variance due to the essence of contrastive learning.

2. We propose two methods: group-p shuffling regulation and logit mean pooling, to mitigate the variance on datapoints and variance across distillation teachers respectively.

3. Experimental results demonstrate our proposed method surpasses the distillation baseline and achieves a new SOTA, which illustrates the effectiveness of our proposed methods.

## 2 DistillCSE for Sentence Embeddings

Self-training utilizes a trained model to generate synthetic labels for examples and trains a student model from scratch (Yarowsky, 1995; scu, 1965). This approach is can be implemented by knowledge distillation (Hinton et al., 2015) when the student model has a equal or smaller capacity than the teacher model. Self-training via knowledge distillation involves two main steps in the context of sentence embeddings as follows.

**Step 1: training teacher model** In the initial phase, a teacher model is trained for sentence embeddings. The mainstream method for sentence embeddings is SimCSE (Gao et al., 2021), which incorporates contrastive learning (CL) principles into the learning process. SimCSE maximizes the agreement of samples with its positive instances while pushing away all the in-batch negative examples using the CL loss defined as follows:

$$\ell_{cl} = -\log \frac{e^{\text{sim}(h_i, h_i^+)/\tau}}{\sum_{j=1}^{N} e^{\text{sim}(h_i, h_j^+)/\tau}} \quad (1)$$

where $h_i = f(x_i)$ represents the embedding of sentence $x_i$ generated by the embedding model $f(\cdot)$, and $h_i^+$ denotes the embedding of the positive instance of $x_i$. The function $\text{sim}(\cdot, \cdot)$ calculates the cosine similarity between two embeddings.

**Step 2: distilling the student model** For the second phase, the student model learns from scratch to mimic the output of the teachers. In this paper, we specifically focus on using a homogeneous model structure for both the teacher and student models (i.e. same capacity). To achieve this, we utilize the vanilla knowledge distillation loss (Hinton et al., 2015), which aims to minimize the cross-entropy of the output logits between the teacher and student.

Formally, given a sentence $x_i$ and its corresponding set of in-batch sentence pairs $\{(x_i, x_j)\}_{j=1, j \neq i}^{N}$, distillation minimizes the cross entropy loss between the teacher distribution $q$ and the student distribution $p$ according to the following objective:

$$\ell_{distill} = -\sum_{i=1}^{N} \sum_{j \neq i, j=1}^{N} q(t_{i,j}) \log p(s_{i,j})$$

$$p(s_{i,j}) = \frac{e^{s_{i,j}/\tau_s}}{\sum_{k=1, k \neq i}^{N} e^{s_{i,k}/\tau_s}} \quad (2)$$

$$q(t_{i,j}) = \frac{e^{t_{i,j}/\tau_t}}{\sum_{k=1, k \neq i}^{N} e^{t_{i,k}/\tau_t}}$$

Here, $s_{i,j}$ and $t_{i,j}$ represent the cosine similarity logits of sentence pairs calculated by the student and teacher models, respectively. $\tau_s$ and $\tau_t$ denote the corresponding temperatures for the student and teacher models, and $N$ denotes the number of sentences within a mini-batch.

In general, the student model is commonly trained jointly with the teacher model's training objective. Hence, the objective for self-training on sentence embeddings combines both objectives using a trade-off factor $\lambda$:

$$\ell = \ell_{\text{cl}} + \lambda \ell_{\text{distill}} \tag{3}$$

**Iterative self-training** Since single-round self-training enhances the student model performance beyond the baseline teacher, a natural assumption is to employ the student as the teacher for the next round of distillation. Consequently, the next-round student could be further improved, which is widely supported by almost all existing findings (Allen-Zhu and Li, 2023; Mobahi et al., 2020). More specifically, iterative self-training employs the $r$-th round student as the $r + 1$-th round teacher:

$$q^{r+1}(t_{i,j}) \leftarrow p^r(s_{i,j}) \tag{4}$$

**Vanilla distillation tends to overfitting** However, our preliminary experiments on vanilla distillation does not obtain significant improvements, which is in line with the finding in Gao et al. (2023). In particular, our further analysis shows that there is a large gap between the loss of the student model on the training and testing sets (See Table 9), which indicates the student model overfits the training corpus. Therefore, we first explore the cause of the overfitting issue and try to tackle such a problem in the following sections.

## 3 High Variance Issues and Solutions

In this section, we will first point out the high variance from the logits that causes the overfitting for knowledge distillation in contrastive learning (i.e., Step 2 in DistillCSE framework) and then propose two simple yet effective solutions to mitigate such issues.

### 3.1 Issues about High Variance of Logits

Conventional self-training involves utilizing teacher information through the equation: $p(y_l|x_i) = e^{w_l^\top h_i} / \sum_k e^{w_k^\top h_i}$, where $w_k$ represents an entry of learnable parameter matrix $W$, and $y_l$ is the task related label. For instance, in language generation tasks, $W$ is the vocabulary embedding matrix and $y_l$ is the token, and in classification tasks, $W$ represents the weighting matrix and $y_l$ is the label in the classifier. Since the embedding $h_i$ is solely related to a single data example $x_i$, the

logits (i.e. each element's magnitude of embedding vector $h_i$) are the 1st-order logits with respect to the random data sample.

However, in sentence embedding settings, we utilize the cosine similarity $t_{i,j}$ for distillation, which is a production of sentence embeddings from a sentence pair $(x_i, x_j)$: $t_{i,j} = h_i^\top h_j$ (assuming both $h_i$ and $h_j$ are $l_2$ normalized). Therefore, the magnitude of logits $t_{i,j}$ depends on both $x_i$ and $x_j$, and thereby $t_{i,j}$ represents the 2nd-order logits with respect to the random data sample. However, since $(x_i, x_j)$ is randomly paired during the training process, this randomness induces two potential issues.

**Variance of logits w.r.t data points** The testing set of STS tasks includes a wide range of sentence pairs with varying degrees of semantic relationship, ranging from strong to weak. However, training samples are randomly gathered from the corpus to form a batch, and thereby the sentences within each batch are completely unrelated. In simpler terms, the level of similarity between the unsupervised training corpus and the testing set may differ significantly from each other.

To demonstrate such distinction, we compare the sample similarity logits within a batch and the similarity logits within STS-B development set. We assess the magnitude of the logit values by sorting them in descending order, as shown in Fig. 1. The distribution of 1st-order embeddings appears similar between the training and testing sets while the 2nd-order similarity logit distribution differs significantly: logits in training sentences exhibit greater concentration, whereas the logits for testing sentences are more moderate and uniform.

We also quantify such a difference in Table 1 by comparing the KL divergence between the training and testing sets. The results reveal that there is not much difference in the 1st-order embedding logits, whereas the 2nd-order similarity logits display significant variation. Consequently, there is a considerable distribution discrepancy in the similarity logits $t_{i,j}$, which may lead to overfitting.

**Variance of logits w.r.t teacher models** Variance may also comes from the teacher model. Since $t_{i,j}$ represents the second-order logits for data samples, its variance theoretically corresponds to the second order of the variance in embeddings. In other words, the variance from teacher embeddings is magnified when transformed into similarity logits.

Table 2 illustrates that the variance in similarity

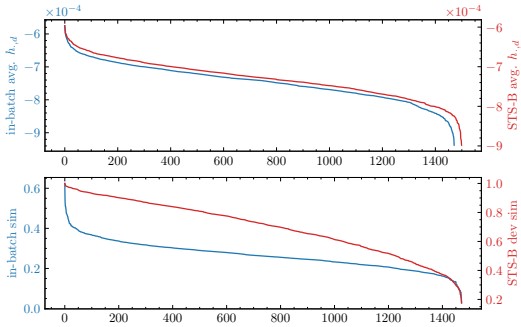

Figure 1: Sharpness of logits between in-batch sentences and STS-B development set.

| Datapoint | $\frac{1}{D}\sum_{d=1}^{D} h_{\cdot,d}$ | | $t_{i,j}$ | |
|---|---|---|---|---|
| | mean | std. | mean | std. |
| Training | -7.43E-04 | 5.25E-05 | 2.67E-01 | 6.40E-02 |
| Testing | -7.29E-04 | 4.61E-05 | 6.96E-01 | 1.81E-01 |
| KL div. | 0.0037 | | 3.8794 | |

Table 1: KL divergence on datapoints between training and testing corpus separately measured on embeddings and logits. $d$ represents embedding dimension and $D$ is the total number of embedding dimensions. (i.e. for base model $D$=768)

logits is significantly greater than the variance in sentence embeddings. Additionally, the differential entropy for similarity logits is even higher compared to embeddings. Moreover, we analyze the impact of this magnified variance: we consider the top 12 logits and calculate their Spearman score across teachers. We find that the Spearman score is only 48.56% for 16 SimCSE teachers across 100 mini-batches. This clearly shows that the variance in embeddings severely disrupts the similarity logits across teachers.

In conclusion, the variance comes from two perspectives: variance on data points and variance across teachers. In other words, the predicted scores for different input samples and teachers vary significantly, which can have implications for the

| | Variable | std. | Diff. Entropy[1] |
|---|---|---|---|
| 1st order | $\frac{1}{D}\sum_{d=1}^{D} h_{\cdot,d}$ | 2.177e-5 | -9.3160 |
| 2nd order | $t_{i,j}$ | 0.0406 | -1.7853 |

Table 2: Average standard deviation and differential entropy for embeddings and similarity logits $t_{i,j}$ across 16 single teachers. 1: Differential entropy can be negative since the logarithm of the PDF can be negative, resulting in a negative value for the integral.

training of the student model. Therefore, we try to improve the learning process by proposing two strategies to mitigate the issue.

## 3.2 Regularization by Shuffling Logits

For the first issue, inspired by the dropout (Srivastava et al., 2014) technique where dropout noise is introduced on the training set to prevent overfitting, we could introduce an outsourced noise into distillation logits to alleviate the overfitting issue caused by high variance. Therefore, a possible solution is to shuffle the teacher logits at a certain interval, which introduces a shuffling noise. This approach helps mitigate overfitting within the interval while preserving valuable information outside such an interval.

We conducted a series of experiments to verify our hypothesis. Specifically, we categorized the logits into five groups by their magnitudes (i.e. model confidence), and for each group, we randomly shuffled the teacher logits during the distillation process. The experimental results are presented in Table 3. It demonstrates that shuffling a subset of the teacher logits effectively addresses the overfitting problem and consequently enhances the performance during testing, especially when the top 12 logits are shuffled. Such results provide evidence that shuffling logits is an effective strategy for mitigating the overfitting issue.

**Group-P shuffling** To further refine the shuffling regulation, we propose Group-P shuffling strategy, which adaptively divides logits into groups and conducts shuffling within each group. Formally, given a sequence of similarity logits for in-batch sentence pairs $T_i = \{t_{i,j}\}_{j=1, j\neq i}^N$, we first compute the logits probability, and its corresponding sorted cumulative probability distribution $G(t_{i,j})$:

$$G(t_{i,j}) = \sum_{t_{i,k} \geq t_{i,j}} \frac{e^{t_{i,k}}}{\sum_{l=1, l\neq i}^N e^{t_{i,l}}} \quad (5)$$

where, the cumulative distribution $G(t_{i,j})$ is the summation of all logits probability that are larger than $t_{i,j}$. Then, we divide $G$ by a probability interval $p$, i.e. $\{p, 2p, \cdots, 1\}$, and finally we randomly shuffle the similarity logits, whose $G(t_{i,j})$ is within the same interval,

$$\hat{T}_i = \texttt{Shuffle}(T_i, \{G(t_{i,j})\}, p) \quad (6)$$

where $p$ is the hyperparameter to trade-off the group size. Large $p$ increase group size for shuffling and prevents the overfitting to teacher logits,

| Methods | STS12 | STS13 | STS14 | STS15 | STS16 | STS-B | SICK-R | Avg. |
|---|---|---|---|---|---|---|---|---|
| SimCSE + Vanilla Distill | 70.85 | 83.49 | 74.84 | 81.52 | 78.19 | 78.60 | 71.69 | 77.03 |
| + Shuffle top 1-12 | **71.70** | 83.37 | **75.62** | **82.28** | **79.23** | **79.65** | **73.09** | **77.85** |
| + Shuffle top 13-24 | 71.50 | 83.75 | 75.34 | 81.83 | 78.69 | 78.77 | 71.74 | 77.37 |
| + Shuffle top 25-36 | 71.30 | 83.67 | 74.94 | 81.92 | 78.23 | 78.50 | 72.21 | 77.25 |
| + Shuffle top 37-48 | 71.51 | **83.78** | 75.34 | 81.82 | 78.67 | 78.75 | 71.66 | 77.36 |
| + Shuffle top 49-63 | 71.23 | 83.66 | 74.96 | 81.97 | 78.21 | 78.50 | 72.17 | 77.24 |

Table 3: Performance of distillation logits using single SimCSE checkpoint under 64 batch size.

while small $p$ reduce the group size and encourage the reliance on the teacher.

### 3.3 Averaging Logits from Multiple Teacher Components

For the second issue, the logits variance across teacher models could be reduced by multiple teacher components: According to the *Central Limit Theorem* (Fischer), the $M$ samples' average converges to its expectation with $1/M$ variance of its original variance. Therefore, we could simply use the following mean sampling to generate the teacher logits $t_{i,j}$:

$$t_{i,j} = \frac{1}{M} \sum_{m=1}^{M} t_{i,j}^m \qquad (7)$$

where $M$ is the total number of teachers, and $t_{i,j}^m$ is the output similarity logit from the $m$-th teacher.

In summary, we propose our DistillCSE distillation method: We still employ the Eq. 2 as the distillation objective. However, we reduce the variance across teachers by averaging logits from multiple teacher components and regulate the student model by group-p shuffling for teacher logits.

## 4 Experiment

### 4.1 Setups

**Baselines** We compare several sentence representation methods on STS tasks, which include GloVe embeddings (Pennington et al., 2014), Skipthought (Kiros et al., 2015), BERT embeddings with pooling aggregation (Devlin et al., 2019), BERT-Flow (Li et al., 2020), and BERT-Whitening (Su et al., 2021). We also compare with several recently proposed CL-based sentence representation methods: ISBERT (Zhang et al., 2020), CT-BERT (Carlsson et al., 2021), ConSERT (Yan et al., 2021), together with the current mainstream SimCSE (Gao et al., 2021) and current SOTA DiffCSE (Chuang et al., 2022). We also conduct the vanilla distillation baseline by ourselves, which leverages the

Eq. 3 to jointly conduct CL and distillation learning from the SimCSE teacher.

**Dataset** We use the default one million randomly sampled sentences from English Wikipedia for unsupervised training, as previous studies (Gao et al., 2021; Chuang et al., 2022; Zhang et al., 2022; Wu et al., 2022) are all conducted on this corpus [1]. We do not conduct any data selection or sampling strategy during the training.

**Evaluation** We evaluate our model on 7 sentence semantic textual similarity (STS) tasks, which includes STS tasks 2012-2016 (Agirre et al., 2012), STS Benchmark (Cer et al., 2017), and SICK-Relatedness (Marelli et al., 2014). We follow SimCSE (Gao et al., 2021) settings of MLP layers and employ MLP on top of [CLS] token representation for training while removing MLP for evaluation. We evaluate the model for every 125 updating steps based on the STS-B development set, without any gradient accumulation. And evaluate the best checkpoint at the final evaluation on test sets.

**Implement details** We conduct the experiments using pre-trained checkpoints from BERT (Devlin et al., 2019) and RoBERTa (Liu et al., 2019) with Huggingface Transformer (Wolf et al., 2020) framework. We employ the current mainstream CL framework SimCSE to train teachers.

During the training, the CL temperature $\tau$, learning batch size, and maximum sequence length are set to $0.05$, $64$, and $32$ respectively, which are the same as the default SimCSE setting. We train the model for 1 epoch. The learning rate for the BERT base model is $3e^{-5}$ while for the large model is $1e^{-5}$, and $1e^{-5}$ for RoBERTa model. The model is optimized by Adam (Kingma and Ba, 2017) optimizer with default settings without any gradient accumulation or momentum CL strategies.

---

[1]https://huggingface.co/datasets/
princeton-nlp/datasets-for-simcse/resolve/main/
wiki1m_for_simcse.txt

| Method | STS12 | STS13 | STS14 | STS15 | STS16 | STS-B | SICK-R | Avg. |
|---|---|---|---|---|---|---|---|---|
| GloVe embeddings (avg.) | 55.14 | 70.66 | 59.73 | 68.25 | 63.66 | 58.02 | 53.76 | 61.32 |
| $BERT_{base}$(first-last avg.) | 39.70 | 59.38 | 49.67 | 66.03 | 66.19 | 53.87 | 62.06 | 56.70 |
| $BERT_{base}$-flow | 58.40 | 67.10 | 60.85 | 75.16 | 71.22 | 68.66 | 64.47 | 66.55 |
| $BERT_{base}$-whitening | 57.83 | 66.90 | 60.90 | 75.08 | 71.31 | 68.24 | 63.73 | 66.28 |
| IS-$BERT_{base}$ | 56.77 | 69.24 | 61.21 | 75.23 | 70.16 | 69.21 | 64.25 | 66.58 |
| CT-$BERT_{base}$ | 61.63 | 76.80 | 68.47 | 77.50 | 76.48 | 74.31 | 69.19 | 72.05 |
| ConSERT-$BERT_{base}$ | 64.64 | 78.49 | 69.07 | 79.72 | 75.95 | 73.97 | 67.31 | 72.74 |
| DiffCSE-$BERT_{base}$ | 72.28 | 84.43 | 76.47 | 83.90 | 80.54 | 80.59 | 71.23 | 78.49 |
| SimCSE-$BERT_{base}$ | 68.40 | 82.41 | 74.38 | 80.91 | 78.56 | 76.85 | 72.23 | 76.25 |
| DCLR-$BERT_{base}$ | 70.81 | 83.73 | 75.11 | 82.56 | 78.44 | 78.31 | 71.59 | 77.22 |
| ArcCSE-$BERT_{base}$ | 72.08 | 84.27 | 76.25 | 82.32 | 79.54 | 79.92 | 72.39 | 78.11 |
| Vanilla-Distill-$BERT_{base}$ | 70.85 | 83.49 | 74.84 | 81.52 | 78.19 | 78.60 | 71.69 | 77.03 |
| *DistillCSE-$BERT_{base}$ | 73.56 | 84.09 | 77.39 | 84.06 | 80.68 | 80.86 | 73.02 | 79.09 |
| *+Teacher Components | 73.14 | 84.36 | 77.05 | 83.64 | 79.94 | 80.21 | 72.15 | 78.64 |
| *+Group-P Shuffling ($p$=0.1) | 72.39 | 83.51 | 75.71 | 82.97 | 78.87 | 79.48 | **73.24** | 78.02 |
| *DistillCSE-$BERT_{base}$ (2nd Round) | **74.54** | **84.51** | **77.67** | **84.87** | **80.70** | **81.48** | 72.16 | **79.42** |
| SimCSE-$BERT_{large}$ | 70.88 | 84.16 | 76.43 | 84.50 | 79.76 | 79.26 | 73.88 | 78.41 |
| DCLR-$BERT_{large}$ | 71.87 | 84.83 | 77.37 | 84.70 | 79.81 | 79.55 | 74.19 | 78.90 |
| ArcCSE-$BERT_{large}$ | 73.17 | 86.19 | 77.90 | 84.97 | 79.43 | 80.45 | 73.50 | 79.37 |
| Vanilla-Distill-$BERT_{large}$ | 72.27 | 85.56 | 77.65 | 84.82 | 80.36 | 80.53 | 75.05 | 79.46 |
| *DistillCSE-$BERT_{large}$ | **75.18** | 86.32 | 78.92 | 85.89 | 81.18 | 81.97 | 75.33 | 80.68 |
| *DistillCSE-$BERT_{large}$(2nd Round) | 75.08 | **86.64** | **79.53** | **86.45** | **81.29** | **82.72** | **76.17** | **81.13** |
| SimCSE-$RoBERTa_{base}$ | 70.16 | 81.77 | 73.24 | 81.36 | 80.65 | 80.22 | 68.56 | 76.57 |
| DCLR-$RoBERTa_{base}$ | 70.01 | 83.08 | 75.09 | 83.66 | 81.06 | 81.86 | 70.33 | 77.87 |
| Vanilla-Distill-$RoBERTa_{base}$ | 71.14 | 82.49 | 73.67 | 81.18 | 81.58 | 81.24 | 68.74 | 77.15 |
| *DistillCSE-$RoBERTa_{base}$ | **71.45** | **83.33** | **75.53** | **83.19** | **82.47** | **82.38** | **69.44** | **78.26** |
| SimCSE-$RoBERTa_{large}$ | 72.86 | 83.99 | 75.62 | 84.77 | 81.80 | 81.98 | 71.26 | 78.90 |
| DCLR-$RoBERTa_{large}$ | 73.09 | 84.57 | 76.13 | 85.15 | 81.99 | 82.35 | 71.80 | 79.30 |
| Vanilla-Distill-$RoBERTa_{large}$ | 72.96 | 84.50 | 76.68 | 85.41 | 82.29 | 82.83 | 71.89 | 79.51 |
| *DistillCSE-$RoBERTa_{large}$ | **74.86** | 85.72 | 78.15 | 86.42 | 83.35 | 84.96 | 73.20 | 80.95 |
| *DistillCSE-$RoBERTa_{large}$(2nd Round) | 73.41 | **85.89** | **78.81** | **86.59** | **83.96** | **84.98** | **74.43** | **81.15** |

Table 4: Experimental results on standard Semantic Textual Similarity tasks. Our proposed method is marked with "*", and Vanilla-Distills are the performance of direct distillation baselines. DistillCSE outperforms Vanilla-Distillation across all types of pre-trained language models with $p < 0.005$.

## 4.2 Main Results

We conduct the experiments with our proposed DistillCSE method for distillation, and the results are shown in Table 4. First, it shows that our proposed DistillCSE-$BERT_{base}$ achieves a 79.09% average Spearman score on STS tasks, which outperforms the distillation baseline Vanilla-Distill-$BERT_{base}$ by 2.08%, and further surpasses its teacher SimCSE-$BERT_{base}$ by 2.87%. Second, we also separately conduct the experiments for shuffling and teacher components. It shows that both proposed strategies yield better performance compared with the distillation baseline, which further demonstrates the effectiveness of our proposed method. Third, combining both strategies finally achieves the best performance across all the baselines, which illustrates that both two strategies are

orthogonal and their gains could be further combined. Finally, we achieve a new SOTA performance on the standard STS tasks across BERT and RoBERTa backbone.

**Discussion on efficiency** Since our proposed method involves multiple teachers for distillation, the main computation overhead arises from inferring the teachers for in-batch negative similarities. To address this, we conduct parallel computation across GPUs. As a result, the overall training overhead is negligible and the training time is comparable to the baseline.

## 4.3 Ablation Study

**Group-P shuffling** We search the $p$ value in set $\{0.05, 0.08, 0.1, 0.12, 0.15\}$ respectively. Table 6 shows the best performance is given by $p = 0.1$.

| Method | STS12 | STS13 | STS14 | STS15 | STS16 | STS-B | SICK-R | Avg. |
|---|---|---|---|---|---|---|---|---|
| SimCSE + Distill | 70.85 | 83.49 | 74.84 | 81.52 | 78.19 | 78.60 | 71.69 | 77.03 |
| + Teacher Comp. | **73.14** | **84.36** | **77.05** | **83.64** | **79.94** | **80.21** | **72.15** | **78.64** |
| Ensemble of Teacher Comp. | 68.85 | 82.46 | 74.07 | 81.21 | 78.95 | 78.92 | 70.66 | 76.45 |

Table 5: Reducing the variance of teacher logits improves the distillation performance while a simple ensemble of teacher components only achieves comparable performance with baseline SimCSE.

**Weightage trade-off factor $\lambda$** Under the distillation baseline settings, we conduct the experiments to search the $\lambda$ in Eq. 3, and the optimal value is 1.

| $p$ | 0.05 | 0.08 | 0.1 | 0.12 | 0.15 |
|---|---|---|---|---|---|
| STS-B | 83.902 | 84.09 | **85.22** | 83.97 | 83.94 |
| $\lambda$ | 0.1 | 0.2 | 0.5 | 1 | 2 |
| STS-B | 83.46 | 83.46 | 83.47 | **83.48** | 83.43 |

Table 6: Searching $p$ and $\lambda$ on STS-B development set.

**Distillation temperatures** Table 7 shows that the distillation performance is robust to the distillation temperatures. Hence, we set the $\tau_s$ and $\tau_t$ to 0.02 and 0.01 respectively, and fix these temperatures across all the experiments.

| $\tau_s \backslash \tau_t$ | 0.05 | 0.02 | 0.01 |
|---|---|---|---|
| 0.05 | 77.01 | 77.03 | 77.07 |
| 0.02 | 76.81 | 76.81 | 77.13 |
| 0.01 | 76.37 | 76.83 | 77.06 |

Table 7: Vanilla-Distill-BERT$_{base}$ baseline average performance on STS tasks with different distillation temperatures.

### 4.4 Empirical Justification on Two Strategies

**Teacher components** We analyze the performance from multiple teacher components in Table 5. It shows that employing teacher components will result in performance increasing to 78.64 Spearman score on average while the ensemble of them only achieves SimCSEs' performance.

**Shuffling logits** We empirically show that group-$p$ shuffling is a regulation that prevents students overfit to teacher model. Fig. 2 shows the loss curve and development set performance during the distillation. It shows that the non-shuffling distillation loss decreases immediately within the first several steps, which implies the model overfits the training corpus. Different from direct distillation, loss for shuffling strategy continues decreasing, which

| Methods | STS-B | Avg. Spearman to | | |
|---|---|---|---|---|
| | | S. T. | O. T. | O. S. |
| non-shuffle | 83.69 | **98.81** | 95.49 | 96.04 |
| shuffle | **83.80** | 98.56 | **95.92** | **96.68** |

Table 8: Distillation model Spearman correlation with other models.

demonstrates shuffling alleviates the overfitting issue. As a consequence, the performance of the shuffling method continues increasing after the non-shuffling method achieves its best performance.

Further, we investigate the distilled student checkpoints in Table 8. We compute the Spearman correlation score on STS-B development set between the student model and: 1) self teacher (S.T.), which is the teacher model used to distill the student; 2) other teachers (O.T.), which are other SimCSEs not used to distillate the current student; 3) other students (O.S.), which are students distilled from other teachers.

First, for the S.T. column, the non-shuffling student has a high correlation with its teacher while the shuffling method reduces the correlation. This indicates shuffling prevents the student from overfitting its own teacher. Second, for the O.T. column, the non-shuffling has a low correlation while the shuffling obtains a high correlation with other

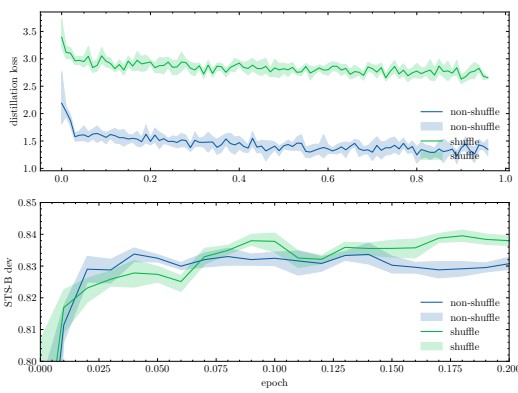

Figure 2: Distillation loss and performance curves between shuffling and non-shuffling strategies.

| Data Split | Vanilla | Group-P | +Logits Avg. |
|---|---|---|---|
| Training | 1.3058 | 2.8715 | 3.1377 |
| Testing | 6.5503 | 6.4410 | 6.2712 |

Table 9: Model's loss value on training and testing set. The large gap between training and testing loss for the vanilla distillation indicates a severe overfitting issue. Our proposed two regulations, i.e. Group-P shuffling and Logits Avg., alleviate the overfitting issue of the vanilla distillation framework.

teachers, which indicates shuffling is helpful to find the common optimum shared across teachers, and it is robust to the specific choice of teacher. Third, the O.S. column demonstrates distillation baseline has a low correlation to others while shuffling increases such a correlation, which indicates shuffling is helpful to achieve the global optimum.

**Two methods alleviate overfitting**  To verify the statement that our proposed two methods alleviate students from overfitting, we measure the student loss value on training and testing datasets in Table 9, respectively. It shows that although vanilla distillation achieves lower training loss, it has a higher loss value on the testing set. While our proposed Group-p shuffling is able to bring the two loss values closer. This phenomenon shows that introducing noise through group-p shuffle has great potential to effectively alleviate the overfitting.

## 5 Related Work

### 5.1 Sentence Embeddings

Early studies for sentence representations inherited the word2vec (Mikolov et al.) ideas: a sentence's contexts shares similar semantic information and such information can be captured by predicting a sentence from its surrounding sentences (Kiros et al., 2015; Hill et al., 2016; Logeswaran and Lee, 2018). Pagliardini et al. (2018) aggregates the n-gram embeddings using a pooling strategy, which achieves a strong result. With the development of large-scale pre-trained language models (Devlin et al., 2019; Liu et al., 2020), sentence representation methods begin to utilize PLMs' strong language representation ability. For example, Reimers and Gurevych (2019) employs siamese network with PLMs for supervised sentence representation, while Li et al. (2020) and Su et al. (2021) apply post-processing on top of PLM's representations.

Recent studies on sentence embeddings are based on the strong baseline of SimCSE (Gao et al., 2021). Under the SimCSE framework, several studies focus on constructing hard contrastive pairs (Zhang et al., 2020; Yan et al., 2021; Giorgi et al., 2021; Kim et al., 2021). Some studies aim to counter the PLMs bias towards sentence representations (Carlsson et al., 2021; Zhou et al., 2022), while others introduce more effective CL framework (Chuang et al., 2022; Clark et al., 2020; Dangovski et al., 2021; Zhang et al., 2022; Xu et al., 2023; Liu et al., 2023).

The development of sentence embeddings has a clear clue: introducing stronger pre-training tasks for PLMs along with CL. The reason is model that achieves better language modeling performance (i.e. token-context alignment) usually has a better ability to capture semantic information and thereby leads to better STS task performance. In contrast to the prior work, this paper aims to study self-training with distillation strategy in sentence embeddings and mainly focuses on the investigation of the factors that affect the model performance. Instead of introducing pre-training tasks for PLMs, we identify the variance from teachers that significantly affect the learning performance and thereby propose methods to tackle those issues.

### 5.2 Contrastive Learning

The importance of contrastive learning (CL) has long been recognized. (Chen et al., 2020; Gidaris et al., 2018; Oord et al., 2018; Wu et al., 2018; Tian et al., 2020a). In NLP research fields, CL is introduced into sentence representations (Giorgi et al., 2021; Wu et al., 2020), text classification (Fang et al., 2020), information extraction (Qin et al., 2021), machine translations (Pan et al., 2021), question answering (Karpukhin et al., 2020) etc.

For example, CL has proven its effectiveness on task-agnostic sentence representations and is further used to improve faithfulness and factuality to generation and summarization. Shu et al. (2021) design rule-based augmentation method on logic-to-text generation, and Cao and Wang (2021) on faithful and factual consistency. In NLP interpretability, Gardner et al. (2020) evaluate local decision boundaries on contrast sets, and Jacovi et al. (2021) develop contrastive explanations for classification models. In contrast to the prior studies that aim to improve performance through CL, we mainly focus on the default CL in a self-training manner, and it is employed as an additive objective

in self-training.

In particular, Gao et al. (2023) study knowledge distillation for contrastive learning on sentence sembeddings similar to our work. However, our work differs theirs in three major aspects. First, our teacher and student are of the same model size whereas they aims to distill a small model from a very large model. Second, we analyze the in-depth reason why vanilla distillation does not work well for contrastive sentence embeddings and propose novel methods to make it successful accordingly. Third, our distillation does not required supervised corpus during the training, making ours more general.

### 5.3 Knowledge Distillation

Knowledge distillation (Hinton et al., 2015) involves training a compact model, often referred to as a student model, to mimic the behavior and knowledge of a larger, more complex model known as the teacher model. It has been successfully applied to various tasks, such as language modeling (Zhuang et al., 2021), text classification (Heinzerling and Strube, 2018; Chia et al., 2019), named entity recognition (Zhou et al., 2021), machine translation (Tan et al., 2019), language generation (Melas-Kyriazi et al., 2019).

Teacher model knowledge guide students in multiple ways during the distillation. Its predictions or soft targets, attention weights, or hidden representations, could all be used to guide the training of the student model. Consequently, the student is provided with stronger training signals from the teacher and achieves even superior performance. For example, Zhuang et al. (2021) directly mimics the output logits on vocabulary while Jiao et al. (2020) utilizes both hidden representations and attention matrix.

Different from those studies, we employ knowledge distillation as an element of our self-training framework. Therefore, we focus on the most fundamental and general form of distillation which only minimize the cross entropy of prediction logits distribution between teacher and students. We use the homogeneous structure model for both the student and teacher model for distillation. Our research mainly focuses on the output logit distribution from the teacher instead of a special distillation framework. Therefore, our method is generic for more advanced distillation technologies.

## 6 Conclusion

In this paper, we propose a self-training with the knowledge distillation framework for contrastive sentence embeddings. We identify that vanilla distillation suffers from severe overfitting issue. The reason for this problem lies in the significant variance of the output logits of the base model in self-training, both among data points and across teachers. Furthermore, reducing the variance will lead to better student performance. Consequently, we propose group-p shuffling to regulate the variance and mean sampling from multiple teacher components to reduce the logit variance. Experimental results on standard benchmarks demonstrate the effectiveness of our proposed method, which yields a new state-of-the-art performance.

## Limitations

This paper identifies variance that significantly affects the distillation learning performance. For variance on data points, a more effective strategy is needed to regulate such variance in logits; For variance across teachers, a more lightweight strategy is needed for teacher components. Besides, the performance of our proposed method could be further improved if a more advanced knowledge distillation framework is introduced.

## Ethics Statement

This study focuses on the self-training methods for contrastive learning sentence embeddings. The proposed objective and methods aim to achieve better performance on general domain tasks. The training corpus is randomly sampled from Wikipedia and benchmark datasets are open source. None of them contain any personally sensitive information; For language models, We employ widely applied pretrained language models, i.e. BERT and RoBERTa, with commonly used contrastive learning and distillation strategies, thereby having no impact on the political, social, or natural environment.

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
