# OpenReview forum: "DistillCSE: Distilled Contrastive Learning for Sentence Embeddings"
_EMNLP/2023/Conference — EMNLP 2023 Findings_

### Official Review · Reviewer_f9XB · 2023-07-27

**Soundness:** 4

**Excitement:**

4: Strong: This paper deepens the understanding of some phenomenon or lowers the barriers to an existing research direction.

**Missing References:**

NA

**Paper Topic And Main Contributions:**

This paper investigates a potential issue with knowledge distillation: the high variance that can arise from the teacher's model trained with Contrastive Learning. To tackle this issue, the authors propose two simple solutions, namely Group-P shuffling and averaging logits from multiple teachers. They show that these two methods do reduce the variance, as well as outperforming existing methods.

**Questions For The Authors:**

On Table 5, which model did you use? Also, the best results don't seem to correspond to any results reported on Table 4, why is this the case? If this is just due to randomness, could you provide std over 5 runs?

**Reasons To Accept:**

The tackled issue is an interesting one, especially when the authors dive into the reasons why high variance occurs when training a model with contrastive learning. The proposed solutions seem relevant to the task, and the overall paper is clear and well organised.

**Reasons To Reject:**

Despite effectively reducing the variance, results seem somewhat limited, only Bert-base being above the baselines with a large margin. Standard deviations are not provided so it is not clear if other results are significant. Moreover, it seems, according to Table 5, that the hyper-parameters tuning for p and $\lambda$ was done on Bert-base, potentially leading to this larger increase in the results for this model only.

Edit: the std have been provided and are small enough for statistical significance, I have updated my scores accordingly.

**Reproducibility:**

3: Could reproduce the results with some difficulty. The settings of parameters are underspecified or subjectively determined; the training/evaluation data are not widely available.

**Reviewer Confidence:**

4: Quite sure. I tried to check the important points carefully. It's unlikely, though conceivable, that I missed something that should affect my ratings.

**Typos Grammar Style And Presentation Improvements:**

On Table 4, DCLR-RoBERTa-base outperforms DistillCSE-RoBERTa-base on both STS15 and SICK-R. Yet, the bold values are not set accordingly.

On Figure 2, the legend is overlapping with the lines, making it hard to read.

---

> ### Author Rebuttal · Authors · 2023-08-29
>
> ### R1: only Bert-base being above the baselines with a large margin; it is not clear if other results are significant; hyper-parameters tuning was done on Bert-base only
>
> **[STD]** We employ vanilla distillation as the target system for the t-test, and models marked with “*” are significantly different from the vanilla distillation model with p <0.005.
> | **Method** | **Avg. STS (confidence interval)** |
> |---|:---:|
> | **SimCSE-BERT-base** | 76.25 (0.1271)* |
> | Vanilla-Distill | 77.03 (0.1165) |
> | *Ours | 79.09 (0.1035)* |
> | *Ours (2nd Round) | 79.42 (0.1035)* |
> | **SimCSE-BERT-large** | 78.41 (0.1164)* |
> | Vanilla-Distill | 79.46 (0.0961) |
> | *Ours | 80.68 (0.1043)* |
> | *Ours (2nd Round) | 81.13 (0.1075)* |
> | **SimCSE-RoBERTa-base** | 76.57 (0.1042)* |
> | Vanilla-Distill | 77.15 (0.1018) |
> | *Ours | 78.26 (0.0969)* |
> | **SimCSE-RoBERT-large** | 78.90 (0.1075)* |
> | Vanilla-Distill | 79.51 (0.0969) |
> | *Ours | 80.95 (0.0888)* |
> | *Ours (2nd Round) | 81.15 (0.0953)* |
>
> **[all types of models outperform the vanilla distillation by p<0.005]**  We provided the performance with t-test confidence interval, it shows that our method on all types of models outperforms both vanilla distillation and SimCSE by a significant margin with respect to the confidence interval value. Specifically, on BERT-large, RoBERTa-base and RoBERTa-large models we increase the performance over vanilla-distill by 1.22, 1.11 and 1.44 respectively, which are substantial margins compared with existing papers.
>
> **[p searching]** Searching specific hyperparameters for each model will surely result in even stronger results for our proposed model. However, we did not do this because the current performance already achieves SOTA performance across models which demonstrates our methods are robust to hyperparameters. Besides, it is a common practice to search the hyperparameters on the BERT-base setting and apply them across all settings in sentence embeddings [1]. Therefore, from a fair comparison perspective, we did not search specific hyperparameters for each model. Tuning the hyperparameters is the last thing to do with the proposed method.
>
>
>
> ### Q1: Model used on Table 5.
>
> The results in Table 5 are surely different from Table 4 because Table 5 are development set results: Table 5 is hyperparameter searching conducted on the STS-B development set, rather than the testing set performance reported in Table 4.Specifically, in Table 5, the p value is searched based on our proposed methods: DistillCSE-BERT-base settings, while the lambda value is searched based on vanilla distillation and we fixed the lambda value across all experiments for a fair comparison between vanilla distillation and our proposed DistillCSE framework.
>
> We provide the 5 runs’ performance of our DistillCSE framework for the first round distillation mean performance (1st row) with the corresponding std (second row) shown below. Our proposed DistillCSE framework outperforms the vanilla distillation method and shares a small performance variance across all types of models.
>
> However, since the second round of distillation requires different first-round distillation checkpoints, which would be a total of 64 checkpoints. We could not finish the experiments in time, but we will add them in the later version.
>
>
>
> | **Methods** | **STS12** | **STS13** | **STS14** | **STS15** | **STS16** | **STS-B** | **SICK-R** | **Avg.** |
> |---|:---:|:---:|:---:|:---:|:---:|:---:|:---:|:---:|
> | **SimCSE-BERT-base** | 68.53 | 80.69 | 73.76 | 81.37 | 78.19 | 77.12 | 70.93 | 75.80 |
> |  | 0.84 | 1.76 | 1.62 | 1.00 | 0.67 | 0.88 | 1.14 | 0.82 |
> | Vanilla | 70.96 | 83.93 | 75.12 | 81.79 | 78.59 | 78.51 | 71.74 | 77.24 |
> |  | 0.11 | 0.65 | 0.52 | 0.46 | 0.35 | 0.14 | 0.09 | 0.25 |
> | Ours | 72.81 | 84.14 | 76.88 | 84.00 | 80.52 | 80.72 | 72.42 | 78.79 |
> |  | 0.35 | 0.37 | 0.03 | 0.24 | 0.11 | 0.11 | 0.07 | 0.11 |
> | **SimCSE-BERT-large** | 71.17 | 84.89 | 76.50 | 84.25 | 79.22 | 79.23 | 74.17 | 78.49 |
> |  | 0.76 | 0.62 | 0.39 | 0.33 | 0.45 | 0.27 | 1.10 | 0.23 |
> | Vanilla | 73.46 | 85.73 | 78.13 | 85.01 | 80.66 | 81.02 | 74.13 | 79.73 |
> |  | 0.76 | 0.42 | 0.32 | 0.24 | 0.14 | 0.50 | 0.55 | 0.10 |
> | Ours | 74.28 | 86.36 | 78.76 | 85.73 | 81.02 | 81.61 | 75.16 | 80.42 |
> |  | 0.58 | 0.22 | 0.21 | 0.26 | 0.17 | 0.21 | 0.21 | 0.17 |
> | **SimCSE-RoBERTa-base** | 69.84 | 81.93 | 74.07 | 82.37 | 80.86 | 81.07 | 68.88 | 77.00 |
> |  | 1.24 | 0.25 | 0.50 | 0.50 | 0.22 | 0.37 | 1.06 | 0.25 |
> | Vanilla | 71.60 | 82.20 | 73.39 | 81.31 | 81.68 | 81.14 | 68.97 | 77.18 |
> |  | 0.64 | 0.42 | 0.40 | 0.18 | 0.13 | 0.15 | 0.32 | 0.04 |
> | Ours | 71.77 | 83.38 | 75.36 | 83.32 | 82.18 | 82.13 | 69.25 | 78.20 |
> |  | 0.41 | 0.23 | 0.20 | 0.24 | 0.17 | 0.17 | 0.40 | 0.10 |
> | **SimCSE-RoBERTa-large** | 71.13 | 83.64 | 75.71 | 84.46 | 81.20 | 82.01 | 70.58 | 78.39 |
> |  | 1.48 | 0.24 | 0.47 | 0.48 | 0.28 | 0.34 | 0.70 | 0.23 |
> | Vanilla | 73.16 | 84.55 | 76.74 | 85.31 | 82.07 | 82.66 | 71.62 | 79.44 |
> |  | 0.28 | 0.06 | 0.08 | 0.15 | 0.32 | 0.25 | 0.39 | 0.10 |
> | Ours | 74.39 | 85.67 | 78.08 | 86.32 | 83.28 | 84.55 | 73.01 | 80.76 |
> |  | 0.34 | 0.31 | 0.17 | 0.19 | 0.18 | 0.41 | 0.22 | 0.14 |
>
>
> ### Reference
> [1] Tianyu Gao, Xingcheng Yao, and Danqi Chen. 2021. SimCSE: Simple Contrastive Learning of Sentence Embeddings. In Proceedings of the 2021 Conference on Empirical Methods in Natural Language Processing, pages 6894–6910, Online and Punta Cana, Dominican Republic. Association for Computational Linguistics.

---

### Official Review · Reviewer_JpHQ · 2023-07-29

**Soundness:** 3

**Excitement:**

3: Ambivalent: It has merits (e.g., it reports state-of-the-art results, the idea is nice), but there are key weaknesses (e.g., it describes incremental work), and it can significantly benefit from another round of revision. However, I won't object to accepting it if my co-reviewers champion it.

**Paper Topic And Main Contributions:**

problem: In self-training through KD  + Contrastive Learning, high variance of teacher’s logits.
solution: Group-P shuffling + teachers’ logit averaging
result: Effective in Semantic Textual Similarity benchmarks + alpha on BERT and RoBERTa base models.


**Questions For The Authors:**

Questions about Reject 1. Reason.
1) Why is it distinguished from many other noising methods for improving over fitting? Especially, are there any specific reasons to categorize logits into confidence-level based groups? To believe that it is safe and understand where to apply it,  we need a deeper explanation  about why the specific inductive bias is positively working beyond just showing its effectiveness in experiments,

**Reasons To Accept:**

1. providing empirical evidence of  the variance issues of second order similarity distribution of logits.
2. method is sufficiently effective in the benchmarks


**Reasons To Reject:**

1. It is not sufficiently demonstrated how the shuffling solves the variance problem.
2. Proposing a DistillCSE as a framework is difficult to  see as a contribution. (Applying CL to representation learning is well-known regularization. Applying CL to Teacher in Self-training framework via KD is simply implemented.)
3. the novelty of raised problem, approach and methods are  weak.

**Reproducibility:**

1: Could not reproduce the results here no matter how hard they tried.

**Reviewer Confidence:**

4: Quite sure. I tried to check the important points carefully. It's unlikely, though conceivable, that I missed something that should affect my ratings.

---

> ### Author Rebuttal · Authors · 2023-08-29
>
> ### R1: How the shuffling solves the variance problem
> **[Problems with vanilla distillation is overfitting]**
> The goal of this work is to address the overfitting problem in vanilla distillation. We noticed that the existing distillation method suffers from overfitting i.e. low training loss but high testing loss.  Large discrepancies between training and test data samples can be the root cause of overfitting on the training set.  Therefore, inspired by dropout technique where dropout noise is introduced on the training set to prevent overfitting, we introduce a regulation on variance (or variance adjustment) into the training set logits via group shuffling, to address overfitting.
>
> To support the above statement about overfitting, we measure the loss on the training and testing sets in the table below. It shows that although vanilla distillation achieves lower training loss, it has a higher loss value on the testing set. While our proposed Group-p shuffling is able to bring the two loss values closer.  This phenomenon shows that introducing noise through group-p shuffle has great potential to effectively alleviate the overfitting problem.
>
> | **Data Split** | **Vanilla** | **Group-p Shuffling** | **+Logits Avg.** |
> |---|:---:|:---:|:---:|
> | Training set | 1.3058 | 2.8715 | 3.1377 |
> | Testing set | 6.5503 | 6.4410 | 6.2712 |
>
>
> The effectiveness of the group-shuffling strategy is also demonstrated and reported in Sec.4.4.  It shows that group shuffling prevents students from overfitting to a particular teacher and encourages students to reach the optimal in global view.
>
> ### R2: Contribution
> **[Contribution]**
> The main contributions are that 1) it points out the overfitting issue for vanilla distillation and explains why the overfitting occurs via measuring the variance for SimCSE; 2) it proposes simple yet effective methods to address the overfitting and finally makes the DistillCSE successful.
>
> Pointing out the vanilla DistillCSE which does not work well is easy, but analyzing its failure in-depth and making the improved DistillCSE successful are solid contributions.
>
> ### R3: novelty of raised problem, approach and methods are weak
> **[Novelty]** DistillCSE, for the first time, studies the distillation sentence embeddings and provides an in-depth analysis of the high variance issue, which can make a significant impact on distillation performance and does not depend on a specific distillation framework. Therefore, our study is generic and significantly contributes to various types of distillation sentence embedding frameworks in future work.
>
> ### Q1: Why shuffling alleviate over fitting?
> **[Group-p Shuffling learns group-level similarity distribution]** The reason is that group-level similarities are more reliable while sentence-level similarities are noisy. This hypothesis is verified in Table 2 where the sentence-level similarities noise std is around 0.04. We provide an example to further illustrate the point: For instance, the similarity distribution of two sentence pairs with similar similarities (e.g. 0.31 and 0.32) are of great noise since std is 0.04, therefore their ground-truth similarities could be 0.31+0.04 and 0.32-0.04, which maybe totally different to the relative scale from teacher’s estimation. Hence **similar similarities** are unreliable due to the noise, and the student tends to overfit to such a noise.  On the contrary, the distribution of sentence pairs with **dissimilar similarities** (e.g. 0.5 and 0.3) is more reliable. Therefore, group-shuffling shuffles the intra-group (similar similarities) distribution while preserving the inter-group (dissimilar similarities) distribution. Consequently, the model learns reliable group-level information while the shuffling noise for similar similarities regulates the overfitting issue.

---

### Official Review · Reviewer_jiJV · 2023-08-04

**Soundness:** 3

**Excitement:**

3: Ambivalent: It has merits (e.g., it reports state-of-the-art results, the idea is nice), but there are key weaknesses (e.g., it describes incremental work), and it can significantly benefit from another round of revision. However, I won't object to accepting it if my co-reviewers champion it.

**Paper Topic And Main Contributions:**

DistillCSE is a sentence embedding learning method that leverages pseudo-labeled data and knowledge distillation. The model is built on three main concepts:

Contrastive Learning: This self-supervised learning technique aims to identify similarities and dissimilarities between sentences. Similar sentences are represented close to each other in the embedding space, while dissimilar ones are kept further apart.

Self-Training Paradigm: The model generates pseudo labels and relearns its weights using both the pseudo labels and correct labels. This allows the utilization of both unlabeled and labeled data.

Knowledge Distillation: In this technique, smaller models are trained to mimic larger models, transferring knowledge from teachers to students in a more compact manner.

Although the original form of DistillCSE does not yield the expected results, the authors propose two solutions. The first is Group-p shuffle, which involves grouping similar sentences, shuffling them, and presenting them to the teacher for pseudo labels. The second solution is to take the teacher's logits for similar sentences and average them.

**Reasons To Accept:**

The idea appears to be innovative, although the actual contribution might be relatively minor. Nevertheless, the authors have tested their model on various tasks, and based on this aspect, I believe their work deserves serious consideration.

**Reasons To Reject:**

The problem with vanilla distillcse is not clear.
They compared their model with BERT but why not Roberta or XLNet.


**Reproducibility:**

N/A: Doesn't apply, since the paper does not include empirical results.

**Reviewer Confidence:**

3: Pretty sure, but there's a chance I missed something. Although I have a good feel for this area in general, I did not carefully check the paper's details, e.g., the math, experimental design, or novelty.

**Typos Grammar Style And Presentation Improvements:**


“Second, it learn a stronger student model through knowledge distillation;” It should be “learns”.
“Thrid” should be Third

---

> ### Author Rebuttal · Authors · 2023-08-29
>
> ### R1: problem with vanilla distillcse is not clear. Why not compare RoBERTa or XLNet?
> **[Vanilla distillation tends to overfit]** The main contributions are that: 1) it points out the overfitting issue for vanilla distillation and explains why the overfitting occurs via measuring the variance for simCSE; 2) it proposes simple yet effective methods to address the overfitting and makes the DistillCSE successful. Therefore, the problem with vanilla distillCSE is the overfitting which is further explained through measuring the variance for simCSE.
>
> To quantify that the vanilla distillCSE suffers from overfitting and our proposed method can alleviate the overfitting, we add new experiments to compare their training loss and testing loss as the following table.
>
> | **Data Splits** | **Vanilla** | **Group-P Shuffling** | **+Logits Avg.** |
> |---|:---:|:---:|:---:|
> | Training Set | 1.3058 | 2.8715 | 3.1377 |
> | Testing Set | 6.5503 | 6.4410 | 6.2712 |
>
>
> As shown in this table, although the vanilla method has low training loss, its testing loss is higher than the group-p shuffling method, which indicates the overfitting to the training set. Moreover, integrating both group-p shuffling and teacher logits averaging yields the lowest testing loss, The large discrepancies between training and test data samples can be the root cause of overfitting on the training set.  The proposed method is able to bring the two loss values closer, and hence reduce the discrepancies.
>
> Consequently, inspired by the dropout technique where dropout noise is introduced on the training set to prevent overfitting, we introduce a regulation on variance (or variance adjustment) into the training set logits via group shuffling, to address overfitting.
>
> We did report the performance of RoBERTa in our main experiments in Table 4. Here, we re-summarize our existing results based on both BERT and RoBERTa from Table 4 as follows:
>
> | **Method** | **Avg. STS (confidence interval)** |
> |---|:---:|
> | **SimCSE-BERT-base** | 76.25 (0.1271)* |
> | Vanilla-Distill | 77.03 (0.1165) |
> | *Ours | 79.09 (0.1035)* |
> | *Ours (2nd Round) | 79.42 (0.1035)* |
> | **SimCSE-BERT-large** | 78.41 (0.1164)* |
> | Vanilla-Distill | 79.46 (0.0961) |
> | *Ours | 80.68 (0.1043)* |
> | *Ours (2nd Round) | 81.13 (0.1075)* |
> | **SimCSE-RoBERTa-base** | 76.57 (0.1042)* |
> | Vanilla-Distill | 77.15 (0.1018) |
> | *Ours | 78.26 (0.0969)* |
> | **SimCSE-RoBERT-large** | 78.90 (0.1075)* |
> | Vanilla-Distill | 79.51 (0.0969) |
> | *Ours | 80.95 (0.0888)* |
> | *Ours (2nd Round) | 81.15 (0.0953)* |
>
> We employ vanilla distillation as the target system for the t-test, and models marked with “*” are significantly different from the vanilla distillation model with p <0.005.
>
> For XLNet, however, we find that SimCSE-XLNet does not converge under the unsupervised SimCSE framework in our experiments. Specifically, we observe that the initial XLNet performance on the STS-B development set is around 50%, and the performance continues to decrease during the training. Finally, it stabilized at around 20% Spearman score. In contrast to unsupervised settings, the supervised SimCSE on top of XLNet achieves on-par performance with unsupervised SimCSE-BERT model.  In addition, to the best of our knowledge, we did not find any existing papers that successfully implement SimCSE on top of XLNet.

---

### Meta-Review · Area_Chair_KQP6 · 2023-09-12

**Recommendation:** 4

**Metareview:**

The paper deals with improving sentence embedding learning. The paper demonstrates that the key issue with knowledge distillation is high variance in the teacher model due to the nature of contrastive loss.  The paper  solves this problem by introducing additional regularization into contrastive training and knowledge distillation. It shows experimentally that the proposed techniques improve BERT performance in a standard sentence similarity benchmark. In rebuttal response the authors show that this method also works with RoBERTa, while experiments with XLNet show that this model does not perform well on this task and cannot be improved within the proposed paradigm.
The reviewers are generally agree in their view on the paper. All raised questions of novelty and meaningful comparison, especially comparison to other regularization techniques.
I would add to concerns raised by the reviewers that the paper does not fit very well into the "Semantics" category, because it does not discuss anything specific to sentence semantics. I would put it under "Machine learning" category, since it discusses a general topic of knowledge distillation with contrastive loss. Thus, I believe the paper might be stronger if the proposed regularization techniques were tested on a broader range of knowledge distillation tasks beyond sentence similarity (and maybe even beyond the NLP realm).

---

### Decision · Program_Chairs · 2023-10-07

**Decision:**

Accept-Findings

**Comment:**

The paper deals with improving sentence embedding learning. The paper demonstrates that the key issue with knowledge distillation is high variance in the teacher model due to the nature of contrastive loss.  The paper  solves this problem by introducing additional regularization into contrastive training and knowledge distillation. It shows experimentally that the proposed techniques improve BERT performance in a standard sentence similarity benchmark. In rebuttal response the authors show that this method also works with RoBERTa, while experiments with XLNet show that this model does not perform well on this task and cannot be improved within the proposed paradigm.
The reviewers are generally agree in their view on the paper. All raised questions of novelty and meaningful comparison, especially comparison to other regularization techniques.
I would add to concerns raised by the reviewers that the paper does not fit very well into the "Semantics" category, because it does not discuss anything specific to sentence semantics. I would put it under "Machine learning" category, since it discusses a general topic of knowledge distillation with contrastive loss. Thus, I believe the paper might be stronger if the proposed regularization techniques were tested on a broader range of knowledge distillation tasks beyond sentence similarity (and maybe even beyond the NLP realm).